# Association between Kihon check list score and geriatric depression among older adults from ORANGE registry

Yu Kume[1], Seongryu Bae[2], Sangyoon Lee[2], Hyuma Makizako[3], Yuriko Matsuzaki-Kihara[4], Ichiro Miyano[5], Hunkyung Kim[6], Hiroyuki Shimada[2], Hidetaka Ota[7] *

1 Department of Occupational Therapy, Graduate School of Medicine, Akita University, Akita, Japan, 2 Center for Gerontology and Social Science, National Center for Geriatrics and Gerontology, Obu, Aichi, Japan, 3 Department of Physical Therapy, School of Health Sciences, Faculty of Medicine, Kagoshima University, Kagoshima, Japan, 4 Department of Rehabilitation, Japan Healthcare College School of Health Sciences, Eniwa, Japan, 5 Department of Public Health, Kochi Medical School, Nankoku-shi, Kochi, Japan, 6 Research Team for Promoting Independence and Mental Health, Tokyo Metropolitan Institute of Gerontology, Tokyo, Japan, 7 Advanced Research Center for Geriatric and Gerontology, Akita University, Akita, Japan

* hidetaka-ota@med.akita-u.ac.jp

**Data Availability Statement:** We cannot publicly provide individual data due to participants' privacy, as specified by the ethics committee. The informed consent obtained does not include a provision for

## Abstract

### Objective

Older adults in Japan are tackling health-related challenges brought by comprehensive geriatric symptoms, such as physical and cognitive problems and social-psychological issues. In this nationwide study, we mainly focused on the Kihon checklist (KCL) as certificated necessity of long-term care for Japanese older adults and investigated whether the KCL score was associated with geriatric depression. In addition, we aimed to identify critical factors that influence the relationship between the KCL score and geriatric depression.

### Methods

This survey was a cross-sectional observational study design, performed from 2013 to 2019. A total of 8,760 participants aged 65 years and over were recruited from five cohorts in Japan, consisting of 6,755 persons in Chubu, 1,328 in Kanto, 481 in Kyushu, 49 in Shikoku and 147 in Tohoku. After obtaining informed consent from each participant, assessments were conducted, and outcomes were evaluated according to the ORANGE protocol. We collected data on demographics, KCL, physical, cognitive and mental evaluations. To clarify the relationship between the KCL and geriatric depression or critical factors, a random intercept model of multi-level models was estimated using individual and provincial variables depending on five cohorts.

### Results

The KCL score was correlated with depression status. Moreover, the results of a random intercept model showed that the KCL score and geriatric depression were associated, and

publicly sharing data. Qualifying researchers may apply to access a minimal dataset by contacting the office of the NCGG-SGS group at https://www.ncgg.go.jp/ri/lab/cgss/department/gerontology/index.html.

**Funding:** This work was supported by the Japan Agency for Medical Research and Development (AMED) (Grant: 18dk0207027h0003) in Japan. AMED had no role in the conduct of the study or interpretation of the results.

**Competing interests:** No authors have competing interests.

its association was affected by provincial factors of slow walking speed, polypharmacy and sex difference.

## Conclusions

These results suggest that provincial factors of low walking performance, polypharmacy and sex difference (female) might be clinically targeted to improve the KCL score in older adults.

## Introduction

A change of lifestyle or a decrease of social participation in older adults have been implicated in the occurrence of geriatric depressive symptoms [1], a decline of physical performance [2, 3] and development of cognitive impairment [4, 5]. Engaging in various social activities is currently considered to be a protective factor against Alzheimer's disease (AD) [6] and geriatric depression [7]. However, despite differences in social and environmental factors among provinces in Japan, examination of critical factors at an individual and provincial level has never been applied in a nationwide survey. Of issues related to epidemiological investigations, recent research has applied multi-level mixed effect model analysis. For example, the influence of dietary environmental factors in twelve provinces of China on obesity among children and adolescents was analyzed using this model [8]. Another study in children has also investigated the prevalence of physical and psychological health-related problems and associated lifestyle factors throughout provinces nationwide [9]. As a result of these studies, the application of multi-level models may reveal critical factors in children's and adolescents' lifestyle. However, thus far, there is insufficient available information to clarify factors in the lifestyle of older adults.

In this study, we aimed to elucidate whether the Kihon checklist (KCL) was related to geriatric depression symptoms, and identify critical provincial factors. The current study utilized the KCL score applying as a certification for long-term care insurance [10]. We herein report a Japanese population-based cross-sectional study of five cohorts in a nationwide clinical registry called the Organized Registration for the Assessment of dementia on Nationwide General consortium toward Effective treatment (ORANGE) [11]. To clarify factors, we performed a random intercept modeling of multi-level models as follows: First, we examined the relationship between KCL score and Geriatric Depression Scale short-form (GDS-15) score at individual level. Second, the relationship between KCL score and provincial factors including sex difference, polypharmacy and usual walking speed was examined in a random intercept modeling. Finally, we identified if the association between KCL score and GDS-15 at the individual level might be affected by usual walking speed at the provincial level [12].

## Methods

### Participants

The Japanese population-based cross-sectional study was conducted from the ORANGE registry at the period from 2013 to 2019. The ORANGE registrants were comprised of 6,755 persons in Chubu province, 1,328 persons in Kanto, 481 persons in Kyushu, 49 persons in Shikoku and 147 persons in Tohoku. The recruitments were performed by publicity papers for citizens of each province. The inclusion criteria were age 65 years and more, having walking ability without personal assistance, and living at home. The exclusion criteria were also dementia, severe hearing or visual impairment, intellectual disability, need for support or care as

certified by the Japanese public long-term care insurance system due to disability, and inability to complete cognitive tests at the baseline assessment. This study was approved by the ethics committee (approval No. 1649) of the Faculty of Medicine, Akita University and was performed in accordance with the Declaration of Helsinki II. After orally explained regarding the aim and methods of this study with the study document, all the participants were signed a consent form for this study to agree to the present study. Evaluations for each cohort were also carried out depending on the ORANGE protocol as the follows;

## Demographic information

Demographic data of age, gender, education, health variables including body mass index (BMI) [kg/m$^2$] and a presence (e.g. each nominal scale [Yes/No] among cohorts) of medical history, such as hypertension, stroke, cardiac disease, diabetes, hyperlipidemia, osteoporosis, respiratory disease, osteoarthrosis (OA), bone fracture history after aged 60, neoplasm, Parkinson's disease (PD), Alzheimer's disease (AD), depression and rheumatoid arthritis (RA).

## Polypharmacy

In Japan, the "Guidelines for Medical Treatment and its Safety in the Elderly 2015" has reported that an increase of occurrence of adverse drug event (ADE) is associated with the number of medications, such as six or more kinds of medications [13]. We recorded the presence of polypharmacy of Japanese classification based on use of six or more medications (polypharmacy-JC). Each participant self-reported the number of medications through the interview by well-trained assistants including medical doctor, public health nurses and physical or occupational therapists.

## Comprehensive Geriatric Assessment (CGA)

Physical, psychological, functional and social status of the participants were comprehensively evaluated using KCL [10] including Q1-25 items (S1 Table). Each score of the KCL indicates difficulty with the activity in the question, and a higher score of the checklist means higher risk of requiring support for each domain. KCL consists of sub-domains (KCL-physical score for physical functions (Q6 to Q10); KCL-nutrition score for nutritional status (Q11, 12); KCL-oral score for oral function (Q13 to Q15); KCL-cognitive score for cognitive function (Q18 to Q20); and KCL-depressive score for depressive mood (Q21 to Q25)). In longitudinal studies investigated predictive ability of incident long-term care insurance certification in the KCL, the fulfillment of ≥10 checked answers of KCL without depressive score (Q1 to Q20) has been documented as older adults who be at the highest risk of becoming certified as needing long-term care. The current study applied the KCL without depression score (Q1 to Q20) [14, 15]. As well as measuring the KCL, well-trained assistants also conducted the Geriatric Depression Scale short-form (GDS-15) [16]. The KCL and GDS-15 were self-administered and then confirmed by face-to-face answer with the assistants for this study to prevent the missing data of each questionnaire.

## Physical and cognitive measurements

Well-trained study assistants carried out the physical and cognitive assessments in the community. Before beginning the investigation, all assistants were given training from the authors to correctly administer the assessments. We measured physical performance consisting of grip strength (kg) and usual walking speed (UWS) (m/s). The grip strength was assessed by the hand of right and left twice, using a Smedley-type handheld dynamometer (GRIP-D; Takei

Ltd., Niigata, Japan). The best value of grip strength was recorded. For measuring the UWS, participants were instructed to walk on a flat and straight surface at a comfortable walking speed, demonstrating the start and end markers of a 5-m walking distance, with a 2-m support walking distance before the start marker and after the end one. The UWS was measured twice and the fast time required was recorded.

The National Center for Geriatrics and Gerontology Functional Assessment Tool (NCGG-FAT) [17] was also completed to assess cognitive domains in the participants, as the follow four subtests;

### 1. Tablet version of word list memory (WM)

The word list memory (WM) test consisted of immediate recognition and delayed recall. The average number of correct answers in three completed trials of the immediate recognition test was scored from 0 to 10. In another delayed recall task, the number of correctly recalled target words was also recorded as a score ranging from 0 to 10. For statistics, the total score of the tasks was applied consequently (from 0 to 20).

### 2. Tablet version of Trail Making Test Version A (TMT-A) and Version B (TMT-B)

In the Trail Making Test Version A (TMT-A) task and Version B (TMT-B) tasks, the required time (seconds) to complete each task within a maximum time of 90 s was recorded.

### 3. Tablet version of Symbol Digit Substitution Task (SDST)

In the SDST matching nine pairs of numbers and symbols, the number of correct numbers within 90 s was scored.

Each cognitive domain of the NCGG-FAT has been standardized cut-off values to definite cognitive impairment (score <1.5 SDs below the age and education-specific means) for an older population-based cohort in communities [18].

### Analyses

After the Kolmogorov-Smirnov test was used to test the normality of variables, the non-parametric tests were applied for analyses of data collected from the ORANGE registry. Spearman correlation analysis was performed to examine the correlation between parameters in all the participants (Table 2).

A random intercept model of the multi-level model (hierarchical linear model, HLM) was performed to examine the association between KCL score and GDS-15. Based on the statistical methodology of previous studies [19, 20], the multi-level model was applied towards the data without the normal distribution. Next, we focused on a relationship between KCL score and variables at provincial level including sex, polypharmacy and UWS. To estimate the available coefficient for the random intercept model, centering within cluster ($_{CWC}$) for variables at individual level and centering at the grand-mean ($_{CGM}$) for those at provincial level were carried out, and the targeted variables of GDS-15 and UWS were converted into GDS-15$_{CWC}$ and UWS$_{CGM}$ for independent variables of the random intercept model [21]. The random intercept modeling was also performed as follows, while referring to Enders et al. (2016) [22]; First, we estimated the null model, as indicated by the intra-class correlation coefficient (ICC) without independent variables to examine the similarity of a dependent variable (e.g., KCL score) for each cohort. Dependent variables to put into the random intercept model were determined according to a result of ICC > 0.05 in KCL's domains (Table 3). Secondly, we estimated the

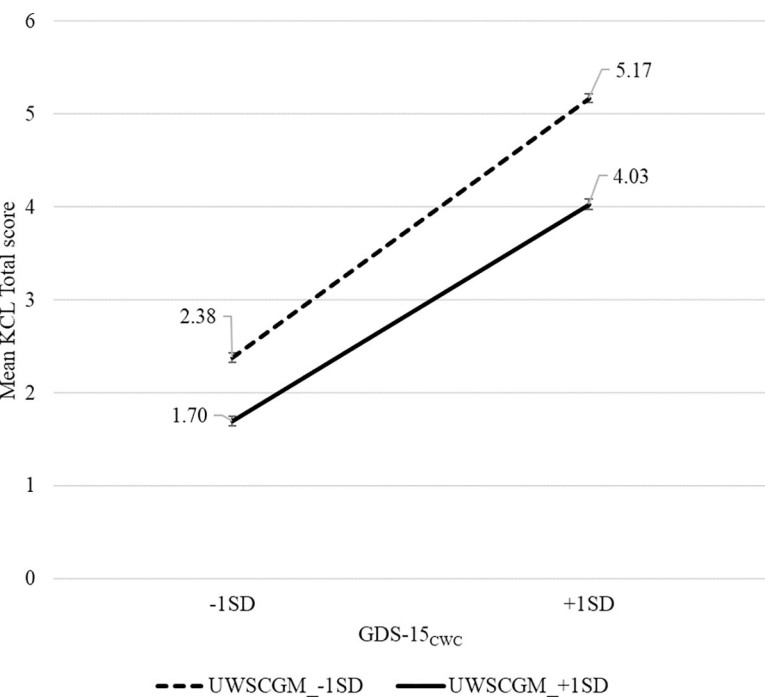

**Fig 1. Main-effect analysis of interaction between GDS-15$_{CWC}$ and UWS$_{CGM}$.** Main-effect analysis, ** $p < 0.01$. The error bar indicates the standard error (SE). KCL, Kihon checklist; UWS, Usual Walking Speed; GDS-15, Geriatric Depression Scale; CGM, centering at the grand-mean; CWC, centering within cluster; SD, standard deviation.

fixed part which input independent variables for random intercept modeling (Tables 4–6). The fixed part was examined with respect to [I] a relationship between KCL score and GDS-15$_{CWC}$ at the individual level and [II] an association between KCL score at individual level and sex difference (female dummy), polypharmacy or UWS$_{CGM}$ at provincial level (S1 Fig). Third, we investigated [III] an interactive effect of GDS-15$_{CWC}$ and UWS$_{CGM}$ toward KCL score at individual level (S1 Fig). Finally, when a significant interactive effect was observed, we examined the main effect of KCL score toward GDS-15$_{CWC}$ at individual level depending on ± 1 standard deviation (SD) of a variable at the provincial level (UWS$_{CGM}$) (Fig 1). Adaptation of each model was assessed by Akaike's Information Criterion (AIC) and Bayesian information criterion (BIC) whose lower values reflect the better model's adaptation. SPSS Version 26.0 for Windows (SPSS Inc., Chicago. IL, USA) was used for analysis, and the level of significance was set at $p = 0.05$.

## Results

A total of 8,760 participants from the ORANGE registry consisted of 6,755 persons in Chubu, 1,328 in Kanto, 481 in Kyushu, 49 in Shikoku and 147 in Tohoku. Table 1 lists demographic data on age, gender, BMI, education, KCL total score and sub-scores, physical, cognitive and mental domains, and medical history for each province.

Next, we analyzed the correlations between KCL scores and physical, cognitive or mental outcomes (Table 2). According to the results of Spearman correlation analysis, KCL total score and sub-scores had a significant correlation with UWS, GS, WM, TMT-A&B, SDST and GDS-15 ($p < 0.01$). KCL total score and KCL without depression score were strongly correlated with the GDS-15 score (KCL total score, $r = 0.44$, $p < 0.01$; KCL without depression score, $r = 0.38$, $p < 0.01$).

**Table 1. Characteristics of participants for each cohort.**

| | Chubu cohort N = 6755 | | Kanto cohort N = 1328 | | Kyushu cohort N = 481 | | Shikoku cohort N = 49 | | Tohoku cohort N = 147 | |
|---|---|---|---|---|---|---|---|---|---|---|
| **Basic information** | Mean | SD | Mean | SD | Mean | SD | Mean | SD | Mean | SD |
| Age (years) | 72.7 | 5.1 | 73.9 | 5.2 | 72.8 | 5.7 | 76.0 | 5.6 | 74.3 | 5.3 |
| Gender (% female) | 52.7% | | 80.0% | | 64.9% | | 87.8% | | 68.7% | |
| BMI (kg/m$^2$) | 23.2 | 3.0 | 22.7 | 3.4 | 23.3 | 3.3 | 25.2 | 3.7 | 24.0 | 3.5 |
| | Median | IQR | Median | IQR | Median | IQR | Median | IQR | Median | IQR |
| Education (year) | 12.0 | 2.0 | 12.0 | 2.0 | 12.0 | 3.0 | 12.0 | 3.0 | 12.0 | 3.0 |
| Medication (n) | 2.0 | 3.0 | 2.0 | 3.0 | 2.0 | 3.0 | 4.0 | 4.5 | 2.0 | 3.0 |
| KCL total score (score) | 3.0 | 4.0 | 3.0 | 4.0 | 3.0 | 3.0 | 6.0 | 4.5 | 3.0 | 4.0 |
| KCL-physical score (score) | 1.0 | 1.0 | 1.0 | 2.0 | 1.0 | 2.0 | 2.0 | 3.0 | 1.0 | 2.0 |
| KCL-nutrition score (score) | 0.0 | 0.0 | 0.0 | 0.0 | 0.0 | 0.0 | 0.0 | 0.0 | 0.0 | 0.0 |
| KCL-oral score (score) | 0.0 | 1.0 | 0.0 | 1.0 | 0.0 | 1.0 | 1.0 | 1.0 | 1.0 | 1.0 |
| KCL without depression score (score) | 2.0 | 3.0 | 2.0 | 3.0 | 3.0 | 2.0 | 5.0 | 3.5 | 3.0 | 4.0 |
| KCL-cognitive score (score) | 0.0 | 1.0 | 0.0 | 1.0 | 0.0 | 1.0 | 0.0 | 1.0 | 0.0 | 1.0 |
| KCL-depression score (score) | 0.0 | 1.0 | 0.0 | 1.0 | 0.0 | 1.0 | 1.0 | 2.0 | 0.0 | 1.0 |
| **Physical domains** | | | | | | | | | | |
| Usual walking speed (m/s) | 1.2 | 0.3 | 1.4 | 0.3 | 1.3 | 0.3 | 1.2 | 0.4 | 1.2 | 0.4 |
| Grip strength (kg) | 26.8 | 12.3 | 22.0 | 6.0 | 24.5 | 10.9 | 20.0 | 6.5 | 23.6 | 7.6 |
| **Cognitive and mental domains** | | | | | | | | | | |
| WM (score) | 12.3 | 4.0 | 13.0 | 4.0 | 13.0 | 4.0 | 9.0 | 4.2 | 12.0 | 4.3 |
| TMT-A (s) | 19 | 5 | 19 | 5 | 19 | 6.5 | 20 | 6.5 | 20 | 6 |
| TMT-B (s) | 33.0 | 14.0 | 34.0 | 15.0 | 33.0 | 16.0 | 38.0 | 19.5 | 35.0 | 16.0 |
| SDST (score) | 46.0 | 12.0 | 45.0 | 12.0 | 45.0 | 13.0 | 38.0 | 14.0 | 42.0 | 12.0 |
| GDS-15 (score) | 2.0 | 2.0 | 2.0 | 3.0 | 2.0 | 3.0 | 5.0 | 6.0 | 2.0 | 3.0 |
| **Medical history** | %Yes | %Miss | %Yes | %Miss | %Yes | %Miss | %Yes | %Miss | %Yes | %Miss |
| Hypertension (%) | 43.9 | 0.00 | 39.9 | 0.00 | 45.3 | 0.00 | 61.2 | 0.00 | 52.8 | 2.04 |
| Stroke (%) | 0.0 | 0.00 | 0.0 | 0.00 | 0.0 | 0.00 | 0.0 | 0.00 | 0.0 | 0.00 |
| Cardiac disease (%) | 17.1 | 0.03 | 13.2 | 0.00 | 8.7 | 0.00 | 22.4 | 0.00 | 24.5 | 0.00 |
| Diabetes (%) | 12.2 | 0.06 | 10.2 | 0.00 | 11.4 | 0.00 | 20.4 | 0.00 | 15.0 | 0.00 |
| Hyperlipidemia (%) | 38.8 | 0.03 | 35.6 | 0.08 | 28.3 | 0.00 | 40.8 | 0.00 | 36.1 | 0.00 |
| Osteoarosis (%) | 11.5 | 0.01 | 22.2 | 0.08 | 17.3 | 0.00 | 26.5 | 0.00 | 24.5 | 0.00 |
| Respiratory disease (%) | 12.6 | 0.03 | 7.9 | 0.00 | 8.9 | 0.00 | 14.3 | 0.00 | 10.2 | 0.00 |
| Osteoarthrosis (%) | 18.8 | 0.07 | 15.3 | 0.00 | 9.6 | 0.00 | 32.7 | 0.00 | 17.1 | 0.68 |
| Bone fracture history after aged 60 (%) | 12.4 | 0.07 | 15.7 | 0.00 | 23.3 | 0.00 | 20.4 | 0.00 | 6.8 | 0.68 |
| Neoplasm (%) | 13.4 | 0.03 | 14.5 | 0.00 | 11.9 | 0.00 | 22.4 | 0.00 | 12.9 | 0.00 |
| Parkinson disease (%) | 0.0 | 0.00 | 0.0 | 0.00 | 0.0 | 0.00 | 0.0 | 0.00 | 0.0 | 0.00 |
| Alzheimer's disease (%) | 0.0 | 0.00 | 0.0 | 0.00 | 0.0 | 0.00 | 0.0 | 0.00 | 0.0 | 0.00 |
| Depression (%) | 0.0 | 0.00 | 0.0 | 0.00 | 0.0 | 0.00 | 0.0 | 0.00 | 0.0 | 0.00 |
| Rheumatoid arthritis (%) | 2.3 | 0.03 | 3.8 | 0.00 | 3.3 | 0.00 | 4.1 | 0.00 | 7.5 | 0.68 |
| Polypharmacy-JC (%) | 13.5 | 0.00 | 11.8 | 0.00 | 15.4 | 0.00 | 24.5 | 0.00 | 13.6 | 0.00 |

%Miss, percent of missing value; SD, standard deviation; IQR, interquartile range; BMI, body mass index; KCL, Kihon Check List; WM, Word list memory; TMT-A, Trail Making Test A version; TMT-B, Trail Making Test B version; SDST, Symbol Digit Substitution Task; GDS-15, Geriatric Depression Scale; Polypharmacy-JC, Polypharmacy based on Japanese classification (e.g. the number of medication is 6 or more).

Table 3 shows the results of ICC (i.e. the null model's estimation) for each KCL score and GDS-15. According to the results of this null model's estimation, KCL total score, KCL-physical score or KCL without depression score of ICC > 0.05 was input as a dependent variable in

**Table 2. Correlation between parameters in all the participants (N = 8,760).**

| | Median | IQR | 1 | 2 | 3 | 4 | 5 | 6 | 7 | 8 | 9 | 10 | 11 | 12 | 13 |
|---|---|---|---|---|---|---|---|---|---|---|---|---|---|---|---|
| 1. KCL total | 3.0 | 4.0 | | | | | | | | | | | | | |
| 2. KCL-physical | 1.0 | 1.0 | .64** | | | | | | | | | | | | |
| 3. KCL-nutrition | 0.0 | 0.0 | .26** | .07** | | | | | | | | | | | |
| 4. KCL-oral | 0.0 | 1.0 | .58** | .23** | .08** | | | | | | | | | | |
| 5. KCL without depression | 2.0 | 3.0 | .96** | .67** | .28** | .60** | | | | | | | | | |
| 6. KCL-cognitive | 0.0 | 1.0 | .48** | .16** | .05** | .21** | .50** | | | | | | | | |
| 7. KCL-depression | 0.0 | 1.0 | .60** | .27** | .09** | .26** | .38** | .22** | | | | | | | |
| 8. UWS | 1.2 | 0.3 | -.22** | -.22** | -.02 | -.09** | -.22** | -.07** | -.13** | | | | | | |
| 9. GS | 25.3 | 11.9 | -.13** | -.23** | -.05** | -.07** | -.11** | .01 | -.13** | .01 | | | | | |
| 10. WM | 12.3 | 4.0 | -.12** | -.09** | -.01 | -.04** | -.11** | -.06** | -.10** | .20** | -.05** | | | | |
| 11. TMT-A | 19.0 | 6.0 | .12** | .12** | .03** | .07** | .11** | .02* | .11** | -.19** | -.08** | -.25** | | | |
| 12. TMT-B | 33.0 | 14.0 | .15** | .13** | .03* | .08** | .14** | .05** | .14** | -.21** | -.07** | -.33** | .49** | | |
| 13. SDST | 46.0 | 12.0 | -.19** | -.19** | -.03* | -.09** | -.17** | -.03** | -.17** | .25** | .22** | .36** | -.53** | -.57** | |
| 14. GDS-15 | 2.0 | 2.0 | .44** | .24** | .08** | .22** | .38** | .22** | .41** | -.13** | -.10** | -.08** | .08** | .11** | -.14** |

*$P < 0.05$

**$P < 0.01$, Spearman rank correlation coefficient.

The values in Table 2 indicate Spearman rank correlation coefficient between parameters.

IQR, interquartile range; KCL, Kihon Check List; UWS, Usual Walking Speed; GS, Grip Strength; WM, word list memory; TMT-A, Trail Making Test A version; TMT-B, Trail Making Test B version; SDST, Symbol Digit Substitution Task; GDS-15, Geriatric Depression Scale.

the following random intercept model. The KCL-depression score (ICC = 0.052) was excluded from dependent variables of the random intercept model due to its being a similar depressive index to GDS-15.

**Table 3. The result of intra-class correlation coefficient (the null model's estimation) among cohorts.**

| Variables | Estimated variance of residuals | Estimated variance of the random intercept | ICC |
|---|---|---|---|
| KCL total score | 7.71 | 0.57 | 0.069 |
| KCL-physical score | 1.13 | 0.06 | 0.054 |
| KCL-nutrition score | 0.18 | NR | NR |
| KCL-oral score | 0.73 | NR | NR |
| KCL without depression score | 4.84 | 0.28 | 0.054 |
| KCL-cognitive score | 0.39 | 0.002 | 0.01 |
| KCL-depression score | 0.99 | 0.05 | 0.052 |
| GDS-15 | 5.36 | 1.00 | 0.16 |

ICC, Intra-class Correlation Coefficient; KCL, Kihon Check List; GDS-15, Geriatric Depression Scale. NR denotes data not reported.

Formula for the ICC, ICC = Intercept [estimate] / (Intercept [estimate] + Residual [estimate]).

To make the first coarse model of the random intercept model (Model I), the independent variables were GDS-15$_{CWC}$ ($\beta = 0.59$, $p < 0.01$). In the second model, GDS-15$_{CWC}$ ($\beta = 0.60$, $p < 0.001$) was significantly associated with KCL total score, with an adjustment variable of cohort mean GDS-15 at provincial level ($\beta = 0.80$, $p < 0.001$). Third, female dummy ($\beta = 0.29$, $p < 0.001$), polypharmacy dummy ($\beta = 0.77$, $p < 0.001$) and UWS$_{CGM}$ ($\beta = -2.18$, $p < 0.001$) at provincial level had significant association with the KCL total score in Model III, as well as GDS-15$_{CWC}$ ($\beta = 0.55$, $p < 0.001$) at individual level. Finally, interactive effects between GDS-15$_{CWC}$ at individual level and UWS$_{CGM}$ at provincial level ($\beta = -0.18$, $p < 0.001$) were significantly observed in Model IV (Table 4, S2 Fig). According to a result of AIC or BIC, the adaptation of Model IV was better than the other models (Table 4, S2 Fig).

Furthermore, the random intercept model towards KCL without depression score demonstrated that GDS-15$_{CWC}$ at individual level or independent variables at provincial level (female dummy, polypharmacy dummy and UWS$_{CGM}$) was significantly associated with the KCL without depression score at individual level (Table 6), as well as a result of KCL-physical score (Table 5). The interactive effects between GDS-15$_{CWC}$ and UWS$_{CGM}$ were significantly documented in each Model IV (Tables 5 and 6).

Finally, we confirmed a main effect of the interaction between GDS-15$_{CWC}$ at individual level and UWS$_{CGM}$ at provincial level, as observed in Model IV of Table 4 (Fig 1). A result of the main effect analysis showed that the group with higher GDS-15$_{CWC}$ (+1 standard deviation [SD]) and lower UWS$_{CGM}$ (-1SD) had significantly higher values of KCL total score ($p < 0.01$).

## Discussion

In this study, we clarified the association between KCL score and GDS-15 in older adults. In addition, a result of the random intercept modeling documented that the association between

**Table 4. Result of random intercept model with a dependent variable of KCL total score.**

| | Model I | | | | Model II | | | | Model III | | | | Model IV | | | |
|---|---|---|---|---|---|---|---|---|---|---|---|---|---|---|---|---|
| | β | | SE | 95%CI | β | | SE | 95%CI | β | | SE | 95%CI | β | | SE | 95%CI |
| **Fixed part** | | | | | | | | | | | | | | | | |
| Intercept | 3.92 | ** | 0.36 | 2.83, 5.00 | 1.50 | *** | 0.32 | 0.87, 2.12 | 2.57 | ** | 0.33 | 1.59, 3.55 | 2.59 | *** | 0.31 | 1.98, 3.20 |
| GDS-15$_{CWC}$ | 0.59 | *** | 0.01 | 0.56, 0.61 | 0.60 | *** | 0.01 | 0.57, 0.62 | 0.55 | *** | 0.01 | 0.53, 0.58 | 0.55 | *** | 0.02 | 0.52, 0.58 |
| Cohort mean GDS-15 | | | | | 0.80 | *** | 0.12 | 0.56, 1.04 | 0.71 | ** | 0.12 | 0.41, 1.02 | 0.69 | *** | 0.12 | 0.46, 0.92 |
| Female dummy | | | | | | | | | 0.29 | *** | 0.05 | 0.19, 0.39 | 0.28 | *** | 0.05 | 0.18, 0.39 |
| Polypharmacy dummy | | | | | | | | | 0.77 | *** | 0.07 | 0.62, 0.92 | 0.75 | *** | 0.07 | 0.61, 0.90 |
| UWS$_{CGM}$ | | | | | | | | | -2.18 | *** | 0.12 | 2.41, -1.95 | -2.18 | *** | 0.12 | -2.41, -1.95 |
| GDS-15$_{CWC}$ × UWS$_{CGM}$ | | | | | | | | | | | | | -0.18 | *** | 0.05 | -0.27, -0.08 |
| **Model's adaptation** | | | | | | | | | | | | | | | | |
| AIC | 40301.51 | | | | 40286.23 | | | | 39793.17 | | | | 39781.39 | | | |
| BIC | 40343.98 | | | | 40335.78 | | | | 39863.95 | | | | 39859.25 | | | |

*$P < 0.05$

**$P < 0.01$

***$P < 0.001$.

Each random intercept model was estimated using Maximum Likelihood (ML).

Polypharmacy-JC, Polypharmacy based on Japanese classification (e.g. the number of medication is 6 or more); UWS, Usual Walking Speed; GDS-15, Geriatric Depression Scale; CWC, centering within cluster; CGM, centering at the grand-mean; SE, standard error; 95%CI, 95% confidence interval; AIC, Akaike's Information Criterion; BIC, Bayesian information criterion

**Table 5. Result of random intercept model with a dependent variable of KCL-physical score.**

| | Model I | | | | Model II | | | | Model III | | | | Model IV | | | |
|---|---|---|---|---|---|---|---|---|---|---|---|---|---|---|---|---|
| | β | | SE | 95%CI | β | | SE | 95%CI | β | | SE | 95%CI | β | | SE | 95%CI |
| **Fixed part** | | | | | | | | | | | | | | | | |
| Intercept | 1.15 | *** | 0.12 | 0.90, 1.39 | 0.29 | * | 0.11 | 0.07, 0.51 | 0.95 | *** | 0.15 | 0.66, 1.24 | 0.90 | *** | 0.14 | 0.62, 1.18 |
| GDS-15$_{CWC}$ | 0.09 | *** | 0.02 | 0.06, 0.11 | 0.11 | *** | 0.01 | 0.09, 0.12 | 0.09 | *** | 0.01 | 0.07, 0.10 | 0.08 | *** | 0.01 | 0.06, 0.10 |
| cohort mean GDS-15 | | | | | 0.28 | *** | 0.05 | 0.19, 0.37 | 0.23 | *** | 0.05 | 0.12, 0.33 | 0.24 | *** | 0.05 | 0.14, 0.34 |
| Female dummy | | | | | | | | | 0.40 | *** | 0.02 | 0.36, 0.44 | 0.40 | *** | 0.02 | 0.35, 0.44 |
| Polypharmacy dummy | | | | | | | | | 0.38 | *** | 0.03 | 0.32, 0.44 | 0.37 | *** | 0.03 | 0.31, 0.43 |
| UWS$_{CGM}$ | | | | | | | | | -1.20 | *** | 0.05 | 1.30, -1.11 | -1.20 | *** | 0.05 | -1.30, -1.11 |
| GDS-15$_{CWC}$ × UWS$_{CGM}$ | | | | | | | | | | | | | -0.06 | ** | 0.02 | -0.10, -0.02 |
| **Model's adaptation** | | | | | | | | | | | | | | | | |
| AIC | 25471.62 | | | | 25459.45 | | | | 24435.28 | | | | 24429.66 | | | |
| BIC | 25514.09 | | | | 25509.00 | | | | 24506.06 | | | | 24507.52 | | | |

*$P < 0.05$

**$P < 0.01$

***$P < 0.001$.

Each random intercept model was estimated using Maximum Likelihood (ML).

Polypharmacy-JC, Polypharmacy based on Japanese classification (e.g. the number of medication is 6 or more); UWS, Usual Walking Speed; GDS-15, Geriatric Depression Scale; CWC, centering within cluster; CGM, centering at the grand-mean; SE, standard error; 95%CI, 95% confidence interval; AIC, Akaike's Information Criterion; BIC, Bayesian information criterion.

KCL score and GDS-15 in older adults was significantly affected by provincial factors including UWS, polypharmacy and sex difference (Tables 4–6, S2 Fig).

After confirming the significant correlation between KCL scores and physical, cognitive or mental outcomes throughout all the samples, we found that KCL total score, KCL physical-

**Table 6. Result of random intercept model with a dependent variable of KCL without depression score (Q1-Q20).**

| | Model I | | | | Model II | | | | Model III | | | | Model IV | | | |
|---|---|---|---|---|---|---|---|---|---|---|---|---|---|---|---|---|
| | β | | SE | 95%CI | β | | SE | 95%CI | β | | SE | 95%CI | β | | SE | 95%CI |
| **Fixed part** | | | | | | | | | | | | | | | | |
| Intercept | 3.19 | *** | 0.25 | 2.69, 3.69 | 1.53 | *** | 0.36 | 0.83, 2.23 | 2.32 | ** | 0.34 | 1.66, 2.98 | 2.34 | *** | 0.34 | 1.69, 3.00 |
| GDS-15$_{CWC}$ | 0.37 | *** | 0.01 | 0.34, 0.39 | 0.38 | *** | 0.01 | 0.35, 0.40 | 0.34 | *** | 0.01 | 0.31, 0.37 | 0.34 | *** | 0.01 | 0.31, 0.37 |
| cohort mean GDS-15 | | | | | 0.56 | *** | 0.13 | 0.31, 0.81 | 0.50 | ** | 0.12 | 0.27, 0.73 | 0.49 | *** | 0.12 | 0.25, 0.72 |
| Female dummy | | | | | | | | | 0.22 | *** | 0.04 | 0.13, 0.30 | 0.22 | *** | 0.04 | 0.13, 0.30 |
| Polypharmacy dummy | | | | | | | | | 0.60 | *** | 0.06 | 0.48, 0.73 | 0.59 | *** | 0.06 | 0.47, 0.72 |
| UWS$_{CGM}$ | | | | | | | | | -1.85 | *** | 0.10 | -2.04, -1.65 | -1.85 | *** | 0.10 | -2.04, -1.65 |
| GDS-15$_{CWC}$ × UWS$_{CGM}$ | | | | | | | | | | | | | -0.09 | ** | 0.04 | -0.17, -0.02 |
| **Model's adaptation** | | | | | | | | | | | | | | | | |
| AIC | 37199.09 | | | | 37190.06 | | | | 36708.56 | | | | 36705.04 | | | |
| BIC | 37241.55 | | | | 37239.61 | | | | 36779.34 | | | | 36782.90 | | | |

*$P < 0.05$

**$P < 0.01$

***$P < 0.001$.

Each random intercept model was estimated using Maximum Likelihood (ML).

Polypharmacy-JC, Polypharmacy based on Japanese classification (e.g. the number of medication is 6 or more); UWS, Usual Walking Speed; GDS-15, Geriatric Depression Scale; CWC, centering within cluster; CGM, centering at the grand-mean; SE, standard error; 95%CI, 95% confidence interval; AIC, Akaike's Information Criterion; BIC, Bayesian information criterion.

score and KCL without depression score were strongly correlated with the GDS-15 score for each province (Table 2). In this way, the relationship between KCL score and GDS-15 became clear. According to the results of ICC in Table 3, we then carried out the random intercept modeling to determine individual or provincial factors that influenced KCL scores in older adults. Tables 4–6 indicated that the association between individual level variables (KCL score and GDS-15$_{CWC}$) was significantly affected by provincial level variables consisting of UWS$_{CGM}$, polypharmacy and female dummy. Particularly, we found that slow walking speed at the provincial level (β = -2.18), polypharmacy-JC (β = 0.75) and female (β = 0.28) were stronger factors on the basis of the association between KCL total score and GDS-15$_{CWC}$ in model IV (Table 4, S2 Fig). From these results, we concluded that these factors at the provincial level (slow usual walking speed, polypharmacy and female) affected the association between KCL score and depression in older adults.

First, our initial aim was to explore the multi-faced state, using the random intercept modeling of multi-level models with a dependent variable of KCL total score or sub-score. When taking a general view of the models across the KCL total, KCL physical and KCL without depression scores, coefficients (β) as reflecting the degree of association with individual or provincial factors were higher in the order of total score (Table 4), Q1—Q20 (Table 6) and physical score (Table 5) of KCL. KCL is generally applied to check early for any decline of physical or mental function in older adults who may have impairment of their living functions [13, 23]. Of the KCL sub-scores, the KCL without depression score, which reflects functioning level in the living environment, tended to have a higher impact on low walking performance, polypharmacy and female as a provincial factor (Table 6) than the physical-score of KCL (Table 5). Therefore, we suggest that KCL without depression score might be a useful tool for screening for a change of functioning level in older community-dwellers. Thus, the comprehensive function in daily life as indexed by the KCL total score has the strongest impact on depression as an individual factor or provincial factors (Table 4), and the KCL without depression score needs to be secondly noted in early screening for depression, low motor function and polypharmacy in older adults (Table 6).

In general, typical linear regression modeling in previous population-based studies disclosed only predictive variables at the individual level. For example, these studies showed an association between the risk of decreased walking speed and number of medications [24] and a negative influence of depression and social activities [25]. Our random intercept model clarified that provincial factors (UWS$_{CGM}$, polypharmacy and female) had the strongest impact on the association between KCL scores and GDS-15$_{CWC}$ at individual level, beyond the limitation of conventional regression methods considering multiple provinces to be one population.

Although the multi-level model methodologically comprises a random intercept model for the fixed effect and a random intercept model for the random effect [17], this study applied random intercept modeling in accordance with research subjects. In short, a result of the random intercept model meant that the KCL total score was significantly associated with GDS-15$_{CWC}$ in older adults and its association between individual variables (the KCL total score and GDS-15$_{CWC}$) was explained by variables at the provincial level (UWS$_{CGM}$, polypharmacy and female (Table 4). In addition, its association of individual factors (the KCL total score and GDS-15$_{CWC}$) were elucidated by significant interactive effects (Interaction [GDS-15$_{CWC}$ × UWS$_{CGM}$], β = -0.18, p < 0.001 in Table 4 and Fig 1). Further analysis of the main effect was performed to clarify if UWS$_{CGM}$ as a provincial factor was important for the association between KCL total score and GDS-15$_{CWC}$ (Fig 1). The results indicated that a provincial factor of UWS$_{CGM}$ had a stronger impact on its association. Therefore, our results not only support the results of conventional regression modeling, but also provide evidence that slow walking

speed, polypharmacy and female as provincial factors could be critical in comprehensive function in daily life and depression in older adults.

Finally, we observed that association between KCL score and geriatric depression might be affected by provincial factors of low walking performance, polypharmacy and female. It has been reported that older adults with no walking outside had less social participation [26], and older adults with poor social participation had an impact on a decline of functional independence [27] and depression [27, 28]. In a longitudinal survey following up 457 Chinese with mean age of 73.6, older adults with depression were associated with polypharmacy and a decline in social activities [29]. As a result, poor functioning states in daily life cause low physical functioning, depression and polypharmacy. Considering these results, we believe that there is a need for preventing a decline of walking performance and inappropriate polypharmacy in older adults, and those changes might be a primary contributor to promoting functioning in daily life and mental health in our community.

## Limitations

Several limitations need to be considered in our study. First, there were different sample sizes and depopulation rate for each province in this study. Especially, the bias of results against larger sample sizes of Shikoku cohort or higher ratio of female of the other cohorts except for Chubu cohort should be considered when interpreting this study's findings. Second, considering the difference in prevalence of polypharmacy among provinces, the relationship between the number of comorbid conditions and the types of medication warrants further examination. Third, one systematic review has reported that KCL is a reliable tool for predicting general frailty and frailty aspects in older adults and there are some cut-offs for the total KCL score (Q1-Q25) to indicate general frailty (≥7 points) or to evaluate prefrail status (three out of four points) and frailty status (seven out of eight points) [30]. Compared with the total KCL score, information obtained about the KCL sub-domains is limited without satisfying validity. To evaluate the appropriateness of the KCL subdomains, the modeling divided the KCL score with sub-domains like KCL-physical score (Q6-Q10) and KCL without depression score (Q1-Q20) could be a new approach in this study. Finally, the random intercept model adjusted for factors at the cluster level evidences that poor functioning states in daily life and depression may be associated with slow walking speed and polypharmacy. Due to a cross-sectional study, this causal relationship cannot be determined, but detailed understanding needs to be validated in a follow-up longitudinal investigation.

## Conclusions

In this study, we applied the random intercept modeling of multi-level analysis in a nationwide survey and clarified critical provincial factors dependent on the association of the KCL score and depression in older community-dwellers. Especially, our study suggested that slow walking speed, polypharmacy and female are related to the state of functioning in daily life and depression in older Japanese adults.

## Supporting information

**S1 Table. Items of the Kihon checklist.**
(DOCX)

**S1 Fig. Procedure of random intercept modeling for research subjects.**
(DOCX)

**S2 Fig. Coefficient (β) and significance for each variable in Model IV with a dependent variable of KCL total score.**
(DOCX)

## Acknowledgments

The authors would like to thank all the participants for their enthusiasm in participating in this project and making this study possible. We would also like to thank Taketo Furuna (Sapporo Medical University School of Health Sciences) for his assistance and management, and all staff at NCGG who provided assistance in performing the assessments.

## Author Contributions

**Conceptualization:** Hiroyuki Shimada.

**Data curation:** Seongryu Bae, Hiroyuki Shimada, Hidetaka Ota.

**Formal analysis:** Seongryu Bae, Hiroyuki Shimada, Hidetaka Ota.

**Funding acquisition:** Hiroyuki Shimada, Hidetaka Ota.

**Investigation:** Sangyoon Lee, Hiroyuki Shimada, Hidetaka Ota.

**Project administration:** Hiroyuki Shimada, Hidetaka Ota.

**Resources:** Sangyoon Lee, Hyuma Makizako, Yuriko Matsuzaki-Kihara, Ichiro Miyano, Hunkyung Kim.

**Software:** Sangyoon Lee, Hiroyuki Shimada.

**Supervision:** Sangyoon Lee, Hyuma Makizako, Hiroyuki Shimada.

**Validation:** Hyuma Makizako, Hiroyuki Shimada.

**Writing – original draft:** Yu Kume, Hidetaka Ota.

**Writing – review & editing:** Yu Kume, Hidetaka Ota.

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
