## [Decision Letter · Decision Letter 0]

18 Jan 2021

PONE-D-20-39864

Low walking performance and polypharmacy affect poor social activities with depression in older adults

PLOS ONE

Dear Dr. Ota,

Thank you for submitting your manuscript to PLOS ONE. After careful consideration, we feel that it has merit but does not fully meet PLOS ONE’s publication criteria as it currently stands. Therefore, we invite you to submit a revised version of the manuscript that addresses the points raised during the review process.

Two experts raised several serious concerns to be clarified.  The authors are recommended to revise carefully.

We look forward to receiving your revised manuscript.

Kind regards,

Tatsuo Shimosawa, M.D., Ph.D.

Academic Editor

PLOS ONE

Journal Requirements:

2. This study was approved by the ethics committee (approval No. 1649) and was performed in accordance with the Declaration of Helsinki II.".   

4. Please include your tables as part of your main manuscript and remove the individual files. Please note that supplementary tables should be uploaded as separate "supporting information" files.

Reviewers' comments:

Reviewer's Responses to Questions

**Comments to the Author**

1. Is the manuscript technically sound, and do the data support the conclusions?

Reviewer #1: Partly

Reviewer #2: Partly

2. Has the statistical analysis been performed appropriately and rigorously? 

Reviewer #1: I Don't Know

Reviewer #2: Yes

3. Have the authors made all data underlying the findings in their manuscript fully available?

Reviewer #1: Yes

Reviewer #2: Yes

4. Is the manuscript presented in an intelligible fashion and written in standard English?

Reviewer #1: Yes

Reviewer #2: Yes

5. Review Comments to the Author

Reviewer #1: First of all, this study regards KCL as an evaluation of social activities, but what is the basis for this? According to the previous articles the “Kihon Checklist” (KCL), a comprehensive evaluation method, was introduced by the Japanese Ministry of Health, Labor and Welfare to identify vulnerable older adults as those at a higher risk of becoming dependent. And it has been reported this KCL is a reliable tool for predicting general frailty and frailty aspects in older adults, but it is not a tool for assessing social activities. According to these various reports the LCL is not a tool to evaluate social activity, although among the 25 items in KCL, 16 and 17 may be related the social isolation. (Geriatr Gerontol Int. 2015 Apr;15(4):518-9.; Geriatr Gerontol Int. 2016 Aug;16(8):893-902.)

Abstract,

“Therefore, we concluded that low walking performance and polypharmacy might play a role in poor social activities with depression in older adults.” Due to a cross-sectional study, this causal relationship cannot be determined.

Methods:

The limitations of this manuscript are the lack of the enough information of the participants as well as sufficient descriptions in the method section.

1. Did these participants recruit from the community or on an outpatient basis?

2. How were the participants recruited?

3. What is the percentage of participation in this study?

4. How did you investigate the number of medicines in the participants? Did you use the self-reported results or did you actually count the number of drugs? Because if it is a self-report, its accuracy is questionable.

5. How was KCL implemented? Is it a self-administered or face-to-face answer?

6. How were grip strength and outpatient high speed measured?

7. How many times have these been measured?

8. What is the cut-off value for each diagnosis of cognitive decline?

Discussion

As can be said for all discussions, it is necessary to recognize that this study is only a cross-sectional analysis and only the relevance has been clarified, and the causal relationship is unknown.

Reviewer #2: The present study investigated whether social activities estimated using the Kihon checklist (KCL) score were affected by depression in older adults estimated using GDS-15. The authors further identified critical factors that affected the association of KCL score with GDS-15 by multi-level mixed-effect model analysis. As the study used multi-level mixed-effect model analysis, which provides a new perspective, it is thought that the study was conducted with sufficient knowledge of statistics, and the results are interesting, but there are some concerns about the analysis method and interpretation of the results.

1. Regarding the ORANGE registry, it is necessary to describe what kind of subjects are recruited and by what method or facility. If the registries are recruited using the same criteria, how do you explain why the five cohorts differ in various indicators?

2. The authors mention in the study limitation, there were different sample sizes for each province. This means that about 77% of participants are from one of five cohorts. If the multivariate analysis is performed only on the data from the Chubu area, it will show that several factors including GDS-15 are independently related to the KCL score, but it will not be possible to find factors that explain the relationship between KCL score and GDS-15. I agree to perform a multi-level mixed-effect model analysis in this study. However, the authors need to explain in more detail in the study limitations why it is statistically meaningful to analyze provincial differences in multi-level mixed-effect model analysis and what the impact on the interpretation should be, given the population composition is so skewed toward one region.

3. Since the present study is a cross-sectional analysis, I don't think it is possible to know whether it is cause or effect. If a multi-level analysis is used, is it possible to determine causality? If the method of analysis could not clarify the causality, the entire description, including conclusions, should be revised to be more cautious.

4. KCL includes questions for depressive mood (Q21 to Q25) as you indicated in line 125. Table 2 shows that the KCL-depression score correlates with many parameters such as KCL total score (ρ=0.60), KCL-life domain (ρ=0.38), and GDS-15 (ρ=0.41). Why did the authors use GDS-15 instead of the KCL-depression score? Why did the authors investigate the effect of GDS-15 on the KCL total score which contains the depression domain in the analysis of multi-level models?

5. As indicated in reference 13 by Satake, et al., the KCL is an easy yes/no self‐reporting questionnaire that has seven domains for comprehensive geriatric assessment. Each domain has a cut-off score. Satake’s study revealed that the total KCL score serves as a screening tool with which to estimate frailty, prefrailty, and robust with cut-offs of 7/8 and 3/4, but did not mention that KCL-total score can be used as a continuous quantity to assess social activities. It needs to be explained, based on previous reports or statistically, that the Spearman rank correlation coefficient can be used to quantitatively process the cumulative score of each domain, and that it is related to the degree of impairment of each domain. In particular, we can imagine that the cumulative score for each domain is correlated with the level of disability sensibly, but I wonder how we can say that the total score is correlated with social activities.

6. Interaction between GDS-15 at individual level and provincial level variable indicates that not only interaction between GDS-15 and usual walking speed but also interaction between GDS-15 and SDST are statistically significant. The authors have not mentioned the interaction between GDS-15 and polypharmacy. How do the readers understand the meaning of interaction between GDS-15 at individual level and provincial level variable in Table 4-6.

7. Is there a consensus that the relevant items for the KCL-life score should be Q1-Q20? If so, please cite it in the reference. Or is it a typographical error since there is no explanation for Q16-Q17?

8. The Geriatric Depression Scale short-form (GDS-15) was performed by trained staff in this study (line 125-126). I understand that GDS15 is a self-check questionnaire, should it be administered by trained staff? I think you used the Japanese version. Whose translation did you use in this study?

9. The citation for Paper 14 is incorrect; "Ageing Ment Health" is not needed.

10. I could not find Supplementary Table 1 (line 119) in the PDF file of the submitted manuscript.

6. PLOS authors have the option to publish the peer review history of their article (what does this mean?). If published, this will include your full peer review and any attached files.

Reviewer #1: No

Reviewer #2: No

---

## [Author Response · Author response to Decision Letter 0]

5 Feb 2021

Academic Editor-in-Chief 'PLOS ONE'

Dear, Professor Tatsuo Shimosawa,

We are most grateful to you for your favorable comments on our manuscript, Article ID number: PONE-D-20-39864: “Low walking performance and polypharmacy affect poor social activities with depression in older adults” which we submitted for consideration for publication in PLOS ONE. We wish to submit a revised version of the manuscript.

According to the suggestions and comments raised by your journal and the reviewers, we have performed additional analyses and re-arranged the manuscript.

We have addressed all the comments of yours as below, the reviewers as indicated on the attached pages, and we hope that the explanations and revision of our work are satisfactory.

We have indicated changes to our manuscript in red.

Data Availability in detail have been described as follows; We cannot publicly provide individual data due to participants' privacy specified by ethics committee. In addition, the informed consent we obtained does not include a provision for publicly sharing data. Qualifying researchers may apply to access a minimal dataset by contacting the office of the NCGG-SGS group at https://www.ncgg.go.jp/cgss/department/cre/.

All the authors contributed to the work and all take responsibility for it. Moreover, none of the work described in the paper has been published or is under consideration for publication elsewhere.

We thank you and all reviewers for the opportunity to revise and resubmit our manuscript to PLOS ONE.

Sincerely yours,

Hidetaka Ota, MD, PhD, Professor, Chief Director,

Advanced Research Center for Geriatric and Gerontology, Akita University,

1-1-1 Hondo, Akita, JAPAN 010-8543

Phone & Fax: +81-18-801-7061/7062

E-mail: hidetaka-ota@med.akita-u.ac.jp

---

## [Decision Letter · Decision Letter 1]

9 Mar 2021

PONE-D-20-39864R1

Low walking performance and polypharmacy are related to poor social activities with depression in older adults

PLOS ONE

Dear Dr. Ota,

Thank you for submitting your manuscript to PLOS ONE. After careful consideration, we feel that it has merit but does not fully meet PLOS ONE’s publication criteria as it currently stands. Therefore, we invite you to submit a revised version of the manuscript that addresses the points raised during the review process.

One original reviewer decline to review your resubmission and I assigned a new reviewer.  Two experts raised serious concern on your manuscript.

We look forward to receiving your revised manuscript.

Kind regards,

Tatsuo Shimosawa, M.D., Ph.D.

Academic Editor

PLOS ONE

Reviewers' comments:

Reviewer's Responses to Questions

**Comments to the Author**

1. If the authors have adequately addressed your comments raised in a previous round of review and you feel that this manuscript is now acceptable for publication, you may indicate that here to bypass the “Comments to the Author” section, enter your conflict of interest statement in the “Confidential to Editor” section, and submit your "Accept" recommendation.

Reviewer #2: (No Response)

Reviewer #3: (No Response)

2. Is the manuscript technically sound, and do the data support the conclusions?

Reviewer #2: Yes

Reviewer #3: No

3. Has the statistical analysis been performed appropriately and rigorously? 

Reviewer #2: Yes

Reviewer #3: No

4. Have the authors made all data underlying the findings in their manuscript fully available?

Reviewer #2: Yes

Reviewer #3: No

5. Is the manuscript presented in an intelligible fashion and written in standard English?

Reviewer #2: Yes

Reviewer #3: No

6. Review Comments to the Author

Reviewer #2: I have pointed out in the previous review that there are some discrepancies between the terminology used by the authors in this paper and what is presented in the original references. The authors should reconsider the terms such as “social activity” in the conclusion of the abstract, and “the KCL-life score”.

The KCL is a screening tool to assess the overall functioning of the elderly. Even if “the KCL-total score” includes indicators that affect social activity, it is not appropriate to use it as an indicator of “social activity” itself. Even though "the KCL-total score" can be an indicator of the overall functioning necessary to maintain "social activity", the more correct terminology should be used with a precise definition in the text. For example, "social capability" is not a commonly used term, and readers are likely to go back to the front of the text to find out what exactly it means. As indicated by another reviewer, the terminology of “social activity” will mislead the readers even the authors add the sentences for the definition. I strongly recommend that the authors change the terminology more correctly.

Regarding the definition or the name of “the KCL-life score” which is the sub-domain of Q1 to Q20, I could not find the word “the KCL-score” in the references that you cited in the responses. Kageyama, et al. used “the 20 Basic Checklist items” as items of Q1 to Q20 and concluded that older residents who corresponded to 10 or more of “the 20 Basic Checklist items” are at the highest risk of becoming certified as needing long-term care. The authors should reconsider the naming about "the sub-domain of Q1 to Q20", instead of “the KCL-life score”.

From the cited references, I understood that “the KCL-life score” can predict prognosis with a certain cutoff value, but I could not find any data that the sum of items classified as frailty is related to prognosis as a continuous variable. I believe that using it as a continuous variable in the analysis in this paper is a new approach, but the validity and limitations of this should be described and discussed in the text.

Regarding the GDS-15, whose translation did you use in this study? Is It the translation by Matsubayashi or one by Sugishita? The authors should clarify it with the correct citation in the reference.

Reviewer #3: The authors investigated the associations between the Kihon checklist (KCL) scores (total, physical and life) with individual- and provincial-level variables related to depressive symptoms, polypharmacy and cognition. The data may provide some interesting information, but the interpretation and conceptual underpinning of the analysis are questionable.

Major concerns:

1. I agree with previous reviewers that the KCL does not measure “social activity” in the usual sense of the phrase. The authors’ response that KCL reflects “comprehensive functions … necessary for engaging in social life” and thus is used “as an indicator of social activity” is flawed. Being alive is also necessary for engaging in social life, which, by the authors’ logic, means being alive can also be used as an indicator of social activity. Clearly no one will accept the latter interpretation, then why should one accept the former? In addition, the references provided by the authors in their response do not “show” KCL reflects the necessity for social activity; rather they use the terms frailty and insurance certification of needs for long-term care.

2. The above point is not to say that one should not examine factors related to KCL scores. It simply says the authors should interpret the data as what the data actually represent.

3. More fundamentally, the authors never really defined what social activity means in the context of their study and never presented a conceptual underlying connection between provincial-level aggregated depressive symptom score, aggregated usual walking speed, and aggregated cognition (trail making test and symbol digit substitution test) with individual-level functioning (physical or life). Because of this lack of clarity, the paper is simply a collection of regression coefficients, which does not improve our understanding of the connection between these variables. For example, why is the individual-level depressive symptom positively correlated with the KCL score but, the provincial-level depressive symptom is negatively correlated with the KCL score? Also because of this lack of clarity, the authors tended to interpret the results in every which way. For example, in the discussion section, the authors first said “… these two factors (slow walking speed and polypharmacy) at the provincial and individual level affected poor social activities with depression,” and then said “the comprehensive state of social activities … has the strongest impact on individual or provincial factors.” Why should individual social activity impact provincial usual walking speed?

Statistical analysis concerns:

1. With 5 clusters, the estimation of between-cluster variation is unlikely to be accurate at all. With so few clusters, the authors attempted to estimate both the random intercept and random slope for depressive symptoms (GDS-15); thus it is not surprising the variance of the random slope could never been truly estimated (no valid standard error) and in two of the three multi-level models the variance of the random intercept cannot be truly estimated either. In fact, with only 5 clusters, the authors used 4 cluster-level covariates in the KCL total and KCL life score models, which should exhaust the degrees of freedom at the cluster-level and make it impossible to estimate either the random intercept or the random slope; and when the authors used 3 cluster-level covariates in the KCL physical score model, they could estimate the random intercept with standard error but not random slope.

2. The authors used the aggregated depressive symptoms, usual walking speed, trail making test and symbol digit substitution test at the provincial level as the factors of interest in Level II of the multi-level model. However, they did not control for other Level II confounding variables such as population size or population composition. In fact, with only 5 clusters, they could not control for other Level II confounders. Based on the descriptive statistics in Table 1, we can see these provinces are quite different in gender composition, which is a major confounding variable at Level II. Differences in aggregated depression, walking speed or cognition could well be due to this gender difference. I suspect if the authors had controlled for this then the other associations would be gone or significantly reduced. This is a major concern.

Minor concerns:

The manuscript would benefit from copyediting by a native English speaker. Some of the following items are language issues.

1. On page 6 line 4, “no normalization by the Kolmogorov-Smirnov test” should be that the Kolmogorov-Smirnov test was used to test the normality of variables.

2. On page 12 Table 3, the column names should be “Estimated variance of residuals” and “Estimated variance of the random intercept.”

3. On page 28 line 18, the sentence should be revised to avoid confusion. For example, it can be changed to “… these two factors at the provincial- (slow walking speed) and individual-level (polypharmacy)…”

4. The phrase “poor social activities with depression” was used in multiple places, including title, abstract, and discussion. This is not accurate. Low walking speed and depressive symptom in Level II and polypharmacy in Level I are associated with KCL, but not KCL with depression.

5. In the abstract, the authors stated the study was “performed from 2015 to 2019,” but in the main text it as “2013 to 2019.”

6. In table 4-6, it will be clearer to put different names for GDS-15, such as GDS-15_CGM (subscript CGM) at the provincial level and GDS-15_CWC (subscript CWC) at the individual level.

7. PLOS authors have the option to publish the peer review history of their article (what does this mean?). If published, this will include your full peer review and any attached files.

Reviewer #2: No

Reviewer #3: No

---

## [Author Response · Author response to Decision Letter 1]

5 Apr 2021

Academic Editor-in-Chief 'PLOS ONE'

Dear, Professor Tatsuo Shimosawa,

We are most grateful to you for your favorable comments on our manuscript, Article ID number:>PONE-D-20-39864R1: “The association between KCL score and GDS-15 was affected by provincial factors of low walking performance, polypharmacy and sex difference” which we submitted for consideration for publication in PLOS ONE. We wish to submit a revised version of the manuscript.

According to the suggestions and comments raised by the reviewers, we have performed additional analyses and re-arranged the manuscript.

We have addressed all the comments of the reviewers as indicated on the attached pages, and we hope that the explanations and revision of our work are satisfactory.

We have indicated changes to our manuscript in red.

All the authors contributed to the work and all take responsibility for it. Moreover, none of the work described in the paper has been published or is under consideration for publication elsewhere.

We thank you and all reviewers for the opportunity to revise and resubmit our manuscript to PLOS ONE.

Sincerely yours,

Hidetaka Ota, MD, PhD, Professor, Chief Director,

Advanced Research Center for Geriatric and Gerontology, Akita University,

1-1-1 Hondo, Akita, JAPAN 010-8543

Phone & Fax: +81-18-801-7061/7062

E-mail: hidetaka-ota@med.akita-u.ac.jp

---

## [Decision Letter · Decision Letter 2]

9 May 2021

PONE-D-20-39864R2

The association between KCL score and GDS-15 was affected by provincial factors of low walking performance, polypharmacy and sex difference

PLOS ONE

Dear Dr. Ota,

Thank you for submitting your manuscript to PLOS ONE. After careful consideration, we feel that it has merit but does not fully meet PLOS ONE’s publication criteria as it currently stands. Therefore, we invite you to submit a revised version of the manuscript that addresses the points raised during the review process.

The statistician raised several concerns to be clarified.

We look forward to receiving your revised manuscript.

Kind regards,

Tatsuo Shimosawa, M.D., Ph.D.

Academic Editor

PLOS ONE

Reviewers' comments:

Reviewer's Responses to Questions

**Comments to the Author**

1. If the authors have adequately addressed your comments raised in a previous round of review and you feel that this manuscript is now acceptable for publication, you may indicate that here to bypass the “Comments to the Author” section, enter your conflict of interest statement in the “Confidential to Editor” section, and submit your "Accept" recommendation.

Reviewer #2: All comments have been addressed

Reviewer #4: (No Response)

2. Is the manuscript technically sound, and do the data support the conclusions?

Reviewer #2: Yes

Reviewer #4: (No Response)

3. Has the statistical analysis been performed appropriately and rigorously? 

Reviewer #2: Yes

Reviewer #4: (No Response)

4. Have the authors made all data underlying the findings in their manuscript fully available?

Reviewer #2: Yes

Reviewer #4: (No Response)

5. Is the manuscript presented in an intelligible fashion and written in standard English?

Reviewer #2: Yes

Reviewer #4: (No Response)

6. Review Comments to the Author

Reviewer #2: (No Response)

Reviewer #4: Important note: This review pertains only to ‘statistical aspects’ of the study and so ‘clinical aspects’ [like medical importance, relevance of the study, ‘clinical significance and implication(s)’ of the whole study, etc.] are to be evaluated [should be assessed] separately/independently. Further please note that any ‘statistical review’ is generally done under the assumption that (such) study specific methodological [as well as execution] issues are perfectly taken care of by the investigator(s). This review is not an exception to that and so does not cover clinical aspects {however, seldom comments are made only if those issues are intimately / scientifically related & intermingle with ‘statistical aspects’ of the study}. Agreed that ‘statistical methods’ are used as just tools here, however, they are vital part of methodology [and so should be given due importance].

COMMENTS: In my opinion, your ABSTRACT is well drafted but assay type. Please note that it is preferable [refer to item 1b of CONSORT checklist 2010: Structured summary of trial design, methods, results, and conclusions] to divide the ABSTRACT with small sections like ‘Objective(s)’, ‘Methods’, ‘Results’, ‘Conclusions’, etc. which is an accepted practice of most of the good/standard journals [including this one]. It will definitely be more informative then, I guess, whatever the article type [I am aware that your Article Type: Research Article] may be.

It is well-known that the Kihon Checklist (KCL) is a self-reported comprehensive health checklist & is used as a screening tool to identify community-dwelling older adults who are vulnerable to frailty and have a higher risk of becoming dependent. Although, according to one famous systematic review ‘KCL is a reliable tool for predicting general frailty and frailty aspects in older adults (Sewo Sampaio PY, Sampaio RA, Yamada M, Arai H. Systematic review of the Kihon Checklist: Is it a reliable assessment of frailty? Geriatr Gerontol Int. 2016 Aug;16(8):893-902. doi: 10.1111/ggi.12833. PMID: 27444395), please check the ‘level of measurement’ of data yielded [KCL scores]. Please refer to lines 11-12 of page 21, ‘Limitations’ section {the use of KCL score as a continuous variable in this study could be a new approach, but it has not been validated.}.

Please note that though the measures/tools used [example: KCL score, Geriatric Depression Scale short-form (GDS-15) score, The word list memory (WM) test, trail making test version A (TMT-A) and version B (TMT-B), Symbol Digit Substitution Task (SDST), etc.] are appropriate, most of them yield data that are in [at the most] ‘ordinal’ level of measurement [and not in ratio level of measurement for sure {as the score two times higher does not indicate presence of that parameter/phenomenon as double (for example, a Visual Analogue Scales VAS score or say ‘depression’ score)}]. Then application of suitable non-parametric test(s) is/are indicated/advisable [even if distribution may be ‘Gaussian’ (i.e. normal)]. Agreed that there is/are no non-parametric test(s)/technique(s) available to be used as alternative in all situation(s) [suitable / most desired/applicable], but should be used whenever/wherever they are available. {Use of KCL score as a continuous variable in this study is surprising and calling it as ‘a new approach’ [line 11 of page 21, ‘Limitations’ section] is even more surprising.}

Therefore, use of non-parametric ‘Spearman correlation coefficient’ [in place of parametric Pearson correlation coefficient (Table 2)] is appreciated but note that ‘multi-level modelling’ is parametric (in my knowledge) which is mostly/mainly used statistical technique for this article. Are the same ‘Conclusions’ can/could be arrived at by some other technique? What is done is not wrong, however, remember that participants of your study are from the ORANGE registry (Please refer to line 2 of page 7. Why no such mention in ‘Methods – Participants? Nothing wrong in being so, but appropriate description is needed). Moreover, note that in a cross-sectional study, causal relationship cannot be determined (Please refer to line 15 of page 21, ‘Limitations’ section).

You may know (I am sure) that “Inferential statistics (i.e. hypothesis testing + estimation of CI) is built on the population model (i.e. the underlying assumption is that there is a population and we are dealing with random sample(s) drawn from that population). Although in clinical trial (involving at least two groups) we do not really deal with random samples (generally a non-probabilistic convenience sampling), ‘allocation’ to treatment groups is ‘randomly’ done which enable us to evoke the population model and we can use inferential statistics safely. But when there is only one group (so that there is no question of random allocation), with ‘non-random’ selection, it may be questionable to use inferential statistics even if you have two measurement sets as ‘pre-post’.” For a pilot study it is alright to have ‘single-arm design’, or when that is the only possibility’, however, it is very essential to keep the limitations in mind while interpreting results.

Part of ‘Conclusions’ section (lines 23-25 of page 21) that “It is possible that these factors may affect the independence of older adults not just in Japan, but in other aging countries” may be correct but is not from this study (your opinion, might have learned/realized during carrying out this study) and how can it be in ‘Conclusions’ section? Title of this article which is “The association between KCL score and GDS-15 was affected by provincial factors of low walking performance, polypharmacy and sex difference”, however, in my opinion, needs to be changed [especially wording, just my opinion and may be considered only if felt worthy].

7. PLOS authors have the option to publish the peer review history of their article (what does this mean?). If published, this will include your full peer review and any attached files.

Reviewer #2: No

Reviewer #4: No

---

## [Author Response · Author response to Decision Letter 2]

13 May 2021

Academic Editor-in-Chief 'PLOS ONE'

Dear, Professor Tatsuo Shimosawa,

We are most grateful to you for your favorable comments on our manuscript, Article ID number:>PONE-D-20-39864R2: “The association between KCL score and GDS-15 was affected by provincial factors of low walking performance, polypharmacy and sex difference” which we submitted for consideration for publication in PLOS ONE. We wish to submit a revised version of the manuscript.

According to the suggestions and comments raised by the reviewer #4, we have performed additional analyses and re-arranged the manuscript.

We have addressed all the comments of the reviewers as indicated on the attached pages, and we hope that the explanations and revision of our work are satisfactory.

We have indicated changes to our manuscript in red.

All the authors contributed to the work and all take responsibility for it. Moreover, none of the work described in the paper has been published or is under consideration for publication elsewhere.

We thank you and all reviewers for the opportunity to revise and resubmit our manuscript to PLOS ONE.

Sincerely yours,

Hidetaka Ota, MD, PhD, Professor, Chief Director,

Advanced Research Center for Geriatric and Gerontology, Akita University,

1-1-1 Hondo, Akita, JAPAN 010-8543

Phone & Fax: +81-18-801-7061/7062

E-mail: hidetaka-ota@med.akita-u.ac.jp

Response to reviewer #4

Thank you very much for supportive reviewer’s comments. According to the reviewer’s suggestion, we have revised our article as follows.

Review Comments to the Author

Reviewer #4: Important note: This review pertains only to ‘statistical aspects’ of the study and so ‘clinical aspects’ [like medical importance, relevance of the study, ‘clinical significance and implication(s)’ of the whole study, etc.] are to be evaluated [should be assessed] separately/independently. Further please note that any ‘statistical review’ is generally done under the assumption that (such) study specific methodological [as well as execution] issues are perfectly taken care of by the investigator(s). This review is not an exception to that and so does not cover clinical aspects {however, seldom comments are made only if those issues are intimately / scientifically related & intermingle with ‘statistical aspects’ of the study}. Agreed that ‘statistical methods’ are used as just tools here, however, they are vital part of methodology [and so should be given due importance].

COMMENTS: In my opinion, your ABSTRACT is well drafted but assay type. Please note that it is preferable [refer to item 1b of CONSORT checklist 2010: Structured summary of trial design, methods, results, and conclusions] to divide the ABSTRACT with small sections like ‘Objective(s)’, ‘Methods’, ‘Results’, ‘Conclusions’, etc. which is an accepted practice of most of the good/standard journals [including this one]. It will definitely be more informative then, I guess, whatever the article type [I am aware that your Article Type: Research Article] may be.

Author’s Response to reviewer’s comments;

Thank you very much for supportive reviewer’s comments. According to the reviewer’s suggestion, we have divided the abstract with small sections of “Objective”, “Methods”, “Results” and “Conclusions”. (Page 2, Abstract)

It is well-known that the Kihon Checklist (KCL) is a self-reported comprehensive health checklist & is used as a screening tool to identify community-dwelling older adults who are vulnerable to frailty and have a higher risk of becoming dependent. Although, according to one famous systematic review ‘KCL is a reliable tool for predicting general frailty and frailty aspects in older adults (Sewo Sampaio PY, Sampaio RA, Yamada M, Arai H. Systematic review of the Kihon Checklist: Is it a reliable assessment of frailty? Geriatr Gerontol Int. 2016 Aug;16(8):893-902. doi: 10.1111/ggi.12833. PMID: 27444395), please check the ‘level of measurement’ of data yielded [KCL scores]. Please refer to lines 11-12 of page 21, ‘Limitations’ section {the use of KCL score as a continuous variable in this study could be a new approach, but it has not been validated.}.

Author’s Response to reviewer’s comments;

Thank you very much for supportive reviewer’s comments. According to reviewer’s suggestion, we have added the reference (21) introduced by the reviewer and have revised the section of “Limitations” as follows; “Third, one systematic review has reported that KCL is a reliable tool for predicting general frailty and frailty aspects in older adults and there are some cut-offs for the total KCL score (Q1-Q25) to indicate general frailty (≥7 points) or to evaluate prefrail status (three out of four points) and frailty status (seven out of eight points) [28]. Compared with the total KCL score, information obtained about the KCL sub-domains are limited without satisfying validity. To evaluate the appropriateness of the KCL subdomains, the modeling divided the KCL score with sub-domains like KCL-physical score (Q6-Q10) and KCL without depression score (Q1-Q20) could be a new approach in this study.” (Page 22, line 11-19).

Please note that though the measures/tools used [example: KCL score, Geriatric Depression Scale short-form (GDS-15) score, The word list memory (WM) test, trail making test version A (TMT-A) and version B (TMT-B), Symbol Digit Substitution Task (SDST), etc.] are appropriate, most of them yield data that are in [at the most] ‘ordinal’ level of measurement [and not in ratio level of measurement for sure {as the score two times higher does not indicate presence of that parameter/phenomenon as double (for example, a Visual Analogue Scales VAS score or say ‘depression’ score)}]. Then application of suitable non-parametric test(s) is/are indicated/advisable [even if distribution may be ‘Gaussian’ (i.e. normal)]. Agreed that there is/are no non-parametric test(s)/technique(s) available to be used as alternative in all situation(s) [suitable / most desired/applicable], but should be used whenever/wherever they are available. {Use of KCL score as a continuous variable in this study is surprising and calling it as ‘a new approach’ [line 11 of page 21, ‘Limitations’ section] is even more surprising.}

Author’s Response to reviewer’s comments;

Thank you very much for supportive reviewer’s comments. As suggested by the reviewer, we have deleted the description of “analysis as a continuous variable” and have added the sentences of “To evaluate the appropriateness of the KCL subdomains, the modeling divided the KCL score with sub-domains like KCL-physical score (Q6-Q10) and KCL without depression score (Q1-Q20) could be a new approach in this study.” (Page 22, line 16-19) in the limitations section.

Therefore, use of non-parametric ‘Spearman correlation coefficient’ [in place of parametric Pearson correlation coefficient (Table 2)] is appreciated but note that ‘multi-level modelling’ is parametric (in my knowledge) which is mostly/mainly used statistical technique for this article. Are the same ‘Conclusions’ can/could be arrived at by some other technique? What is done is not wrong, however, remember that participants of your study are from the ORANGE registry (Please refer to line 2 of page 7. Why no such mention in ‘Methods – Participants? Nothing wrong in being so, but appropriate description is needed). Moreover, note that in a cross-sectional study, causal relationship cannot be determined (Please refer to line 15 of page 21, ‘Limitations’ section).

You may know (I am sure) that “Inferential statistics (i.e. hypothesis testing + estimation of CI) is built on the population model (i.e. the underlying assumption is that there is a population and we are dealing with random sample(s) drawn from that population). Although in clinical trial (involving at least two groups) we do not really deal with random samples (generally a non-probabilistic convenience sampling), ‘allocation’ to treatment groups is ‘randomly’ done which enable us to evoke the population model and we can use inferential statistics safely. But when there is only one group (so that there is no question of random allocation), with ‘non-random’ selection, it may be questionable to use inferential statistics even if you have two measurement sets as ‘pre-post’.” For a pilot study it is alright to have ‘single-arm design’, or when that is the only possibility’, however, it is very essential to keep the limitations in mind while interpreting results.

Author’s Response to reviewer’s comments;

Thank you very much for supportive reviewer’s comments. As suggested by the reviewer, the participants, we have revised the section of “Methods” including the description about “Participants” from the ORANGE registry. (Page 3, line 23-24). According to a result of the normal distribution test, the data collected from the ORANGE registry was analyzed by the non-parametric test. The above explanation has been added into the section of Methods. Based on studies previously reported by Raudenbush & Chan (1993) or Otani, Deng et al. (2020), we understand that multi-level analysis (another name; hierarchical linear model, HLM) can be applied towards the data without the normal distribution. 

Also, we guess that, as commented by the reviewer, the description of “multi-level modelling’ is parametric (in my knowledge) which is mostly/mainly used statistical technique for this article” would correspond to another statistical methodology of “Hierarchical Multiple Regression Analysis：HMRA”. The HMRA is a parametric method and need to be analyzed by the data with the normal distribution. Consequently, we have applied the random intercept modeling of the HLM with the data without the normal distribution, and have added the reference 19, 20, and the explanation of “the multi-level model (hierarchical linear model, HLM)” in the section of Methods-Analyses (page 6, line 9-12).

Reference 19) Raudenbush SW, Chan WS. (1993). Application of a hierarchical linear model to the study of adolescent deviance in an overlapping cohort design. Journal of consulting and clinical psychology. 61:941-951.

Reference 20) Otani K, Deng Y, Herrmann PA, Kurz RS. (2020). Patient Satisfaction, Quality Attributes, and Organizational Characteristics: A Hierarchical Linear Model Approach. J Patient Exp. 7:801-806.

Part of ‘Conclusions’ section (lines 23-25 of page 21) that “It is possible that these factors may affect the independence of older adults not just in Japan, but in other aging countries” may be correct but is not from this study (your opinion, might have learned/realized during carrying out this study) and how can it be in ‘Conclusions’ section? Title of this article which is “The association between KCL score and GDS-15 was affected by provincial factors of low walking performance, polypharmacy and sex difference”, however, in my opinion, needs to be changed [especially wording, just my opinion and may be considered only if felt worthy].

Author’s Response to reviewer’s comments;

Thank you very much for supportive reviewer’s comments. Considering the reviewer’ comment, we have deleted the sentence. 

Additionally, according to the reviewer’s suggestion, a title of our article has been changed revised as follows” Association between Kihon check list score and geriatric depression among older adults from Orange registry”.

---

## [Decision Letter · Decision Letter 3]

21 May 2021

Association between Kihon check list score and geriatric depression among older adults from Orange registry

PONE-D-20-39864R3

Dear Dr. Ota,

We’re pleased to inform you that your manuscript has been judged scientifically suitable for publication and will be formally accepted for publication once it meets all outstanding technical requirements.

Kind regards,

Tatsuo Shimosawa, M.D., Ph.D.

Academic Editor

PLOS ONE

Additional Editor Comments (optional):

Reviewers' comments:

Reviewer's Responses to Questions

**Comments to the Author**

1. If the authors have adequately addressed your comments raised in a previous round of review and you feel that this manuscript is now acceptable for publication, you may indicate that here to bypass the “Comments to the Author” section, enter your conflict of interest statement in the “Confidential to Editor” section, and submit your "Accept" recommendation.

Reviewer #4: All comments have been addressed

2. Is the manuscript technically sound, and do the data support the conclusions?

Reviewer #4: Yes

3. Has the statistical analysis been performed appropriately and rigorously? 

Reviewer #4: Yes

4. Have the authors made all data underlying the findings in their manuscript fully available?

Reviewer #4: Yes

5. Is the manuscript presented in an intelligible fashion and written in standard English?

Reviewer #4: Yes

6. Review Comments to the Author

Reviewer #4: COMMENTS: All the comments made on earlier draft(s) by me (and hopefully by other respected reviewers also) were/are attended positively/adequately, I am fully satisfied and the manuscript is improved a lot. I recommend acceptance.

7. PLOS authors have the option to publish the peer review history of their article (what does this mean?). If published, this will include your full peer review and any attached files.

Reviewer #4: **Yes: **Dr. Sanjeev Sarmukaddam

---

## [Editor Report · Acceptance letter]

27 May 2021

PONE-D-20-39864R3 

Association between Kihon check list score and geriatric depression among older adults from Orange registry 

Dear Dr. Ota:

I'm pleased to inform you that your manuscript has been deemed suitable for publication in PLOS ONE. Congratulations! Your manuscript is now with our production department. 

Kind regards, 

on behalf of

Prof. Tatsuo Shimosawa 

Academic Editor

PLOS ONE